# Vulnerability Analysis to Maximize the Resilience of Power Systems Considering Demand Response and Distributed Generation

**Darin Jairo Mosquera Palacios** [1,*], **Edwin Rivas Trujillo** [1] **and Jesús María López-Lezama** [2]

[1] Research Group Electromagnetic Compatibility and Interference (GCEM), Universidad Distrital Francisco José de Caldas, Bogotá 11021, Colombia; erivas@udistrital.edu.co
[2] Research Group of Efficient Energy Management (GIMEL), Universidad de Antioquia (UdeA), Medellin 050010, Colombia; jmaria.lopez@udea.edu.co
[*] Correspondence: djmosquerap@correo.udistrital.edu.co

**Abstract:** Electric power systems are subject to failures, due to both deliberate and fortuitous events. This paper addresses the first case in which a disruptive agent aims at maximizing the damage to the network (expressed through the total cost of operation), while the system operator takes the necessary measures to mitigate the effects of this attack. The interaction between these two agents is modeled by means of a bi-level optimization problem. On one hand, the disruptive agent is positioned in the upper-level optimization problem and must decide which elements to render out of service (lines and generators), given a limited destructive budget. On the other hand, the system operator, located in a lower-level optimization problem, reacts to the attack by deploying mitigation measures in order to minimize cost overruns in system operation. Based on the aforementioned dynamic, this paper proposes a novel approach to maximize the resiliency of the power system under intentional attacks through the implementation of distributed energy resources (DERs), namely, distributed generation (DG) and demand response (DR). Three metrics are proposed to assess resilience by assigning DERs in islands generated by the destruction of lines and generators. The results obtained in a didactic 5-bus test system and the IEEE RTS-24 bus test system demonstrate the applicability and effectiveness of the proposed methodology.

**Keywords:** demand response; distributed generation; disruptive event; vulnerability analysis; load shedding; resiliency

## 1. Introduction

### 1.1. Motivation

Electric power systems (EPS) play a key role in modern society since the productivity of industries and welfare of communities largely depend on their appropriate functioning. Therefore, power system operators and planners must make efforts to guarantee the quality and continuity of energy supply. Unfortunately, EPS are vulnerable not only to natural events, but also to malicious attacks [1]. These disruptive events bring along high operating costs, due to unforeseen changes in the initial dispatch plan, repair costs of elements such as lines, transformers, and towers, as well as eventual compensations to consumers due to load shedding. Power system operators are in charge of assessing such costs and developing strategies to minimize the impact of eventual outages [2]. Since the costs of outages are high for both the consumers and the network operator, it is necessary to establish strategies before, during and after an attack to mitigate the impact and duration of service interruptions [3]. Due to the fact that transmission networks operate over a wide geographic area, they are particularly susceptible to deliberate physical attacks. In this case, there is a disruptive or malicious agent that wants to cause damage to the system. This agent has access to the system data and has limited economic resources to execute its

objective; therefore, such an agent must determine a set of elements to render out of service with the aim of maximizing damage to the network. At the same time, the system operator must react to the attack to minimize the damage. This attacker–defender dynamic was initially modeled in [3] through a bi-level programming approach. In this case, the authors propose analytical techniques to help mitigate the disruptions to electric power grids caused by terrorist attacks. They also identify critical system components (lines, generators and transformers) by creating disruptive attack plans, assuming limited offensive resources. From the seminal work developed in [3], many models and techniques have been proposed to approach the vulnerability assessment of power systems, bearing in mind intentional attacks; this is also known as the electric grid interdiction problem (EGIP).

### 1.2. Literature Review

The EGIP was initially proposed in [3]. In this case, the objective function of the terrorist and system operator are the maximization and minimization of the load shedding, respectively resulting in a max–min optimization model. In [4], the authors propose a generalization of the EGIP by introducing different objective functions for the terrorist and the system operator. The new model also allows the imposition of constraints on the outer optimization that are functions of both the inner and outer variables. After a given attack, the system operator usually resorts to the use of a new generation to minimize the system load shedding; nonetheless, in [5], the authors propose line switching as an alternative method to protect the system. In this case, a genetic algorithm (GA) is in charge of finding a new system topology that would mitigate the effects of an attack. In the face of a disruptive event, the system operator may also take advantage of distributed energy resources (DERs) to provide an alternative to supply some critical loads while the lost infrastructure is reestablished [6,7]. Distributed generation (DG) and demand response (DR) are types of DERs that increase capacity to effectively transfer power from generators to loads, decrease the power demand on loads through bilateral agreements for voluntary disconnection and minimize load disconnection costs while recovering lost infrastructure [8], which is reflected in the increased flexibility, robustness and reliability of the power system [9].

From the standpoint of game theory, the EGIP can be seen as a leader–follower Stackelberg game in which the leader must anticipate the reaction of the follower. From the standpoint of mathematical programming, a bi-level optimization model is a mathematical problem with equilibrium constraints, which is intrinsically non-linear and non-convex [10]. Therefore, the best way to deal with this type of problem is to resort to a single-level equivalent, which can be done by using the Karush–Kuhn–Tucker optimality conditions and the duality theory [11]. Nonetheless, this can only be applied when the lower-level optimization problem is linear, which in this case, requires a DC approximation of the power flow equations that govern the transmission network. Due to this fact, many works regarding the EGIP adopt linear modeling of the transmission grid. This limitation is overcome in [12], where the authors propose a non-linear modeling of the lower-level optimization problem and deal directly with the bi-level optimization problem through a metaheurstic technique, without needing linear approximations. In [13], the authors also approach the EGIP with non-linear modeling of the network, considering the effect of DR and using a metaheuristic technique.

After a malicious attack is executed in a power system, multiple outages may take place. In this case, it becomes crucial to quickly identify the transmission links that have a limited power transfer capability (these critical interconnections are known as cut-sets). In [14], the authors apply graph theory for analyzing whether a given contingency may result in saturated cut-sets; in this way, situational awareness is arisen and corrective actions can be quickly carried out. Graph theory is also used in [15] within the context of vulnerability analysis. In this case, a cascading failure model is developed based on the continuous temperature evolution process of lines. This method allows characterizing the vulnerability of lines and transmission networks. In [16], the authors use AC modeling of the network and perform a vulnerability analysis, aiming at identifying critical nodes.

A similar approach is proposed in [17], where also critical buses are identified, using a geodesic vulnerability index. In [18], the authors propose a novel measurement to quantify the robustness of power grids under outages by means of a network hierarchy evolution analysis. In [19], a vulnerability assessment of power systems under failures or attacks is carried out based on topological properties. In this case, a particle swarm optimization (PSO) algorithm is proposed to identify critical elements that may trigger cascading failures. A maximum flow-based complex network approach is proposed in [20] to identify critical lines in a power system. The proposed approach consists of two steps. In the first one, the power system is modeled, using graph theory in which the nodes are represented by substations, and the edges correspond to lines and transformers. In this case, the critical outage scenarios are identified, using principal component analysis. In the second step, a topology analysis is implemented through a maximum flow-based network approach.

### 1.3. Contributions and Paper Organization

This paper differentiates from graph theory approaches, such as those in [14,15,19], in the sense that not only buses but also lines and generators may be identified as critical elements. Furthermore, unlike [3–5], AC modeling of the network is implemented, which allows a more realistic approach to the problem. It also differentiates from other EGIP models in the sense that it proposes new resiliency metrics and considers the use of DERs within the options of the system operator to react when facing a malicious attack. To summarize, the main features and contributions of this paper are listed below.

- It complements previous works reported in the specialized literature regarding the solution of the EGIP by considering AC modeling of the problem as well as simultaneous attacks on lines and generators.
- New metrics are proposed for the assessment of power system resiliency under deliberate attacks.
- Enhancement of grid resiliency is proposed by introducing the effect of DERs as a reaction strategy of the system operator.

The rest of this document is organized as follows. Section 2 presents an outline of the EGIP. Section 3 describes the mathematical modeling and solution approach implemented. Section 4 elaborates on the tests and results. Section 5 displays a discussion of results. Finally, Section 6 explains the conclusions of this work.

### 2. Outline of the EGIP

The solution of the EGIP allows identifying the critical elements (lines, transformers and generators) that, if attacked, would cause the greatest damage to the system in terms of costs due to mandatory load disconnection (load shedding) and the use of generation out of merit (expensive generation plans) that were not initially considered in the grid operation [21]. The vulnerability analysis has as input data the normal operating conditions, the evaluation of the resources allocated by the network operator to protect each element, the monitoring of voluntary disconnection contracts, the resources of the disruptive agent and the costs of load shedding at each of the nodes. In this case, there is an interaction between the disruptive agent and the network operator. There is also a dependency between the decision variables of each of them. This dependency is modeled by means of a two-level optimization problem as depicted in Figure 1. The disruptive agent conceives an interdiction vector (action), acknowledging the reaction of the system operator (reaction). Once an attack plan is executed, the system operator runs an optimal power flow to minimize system costs. This optimization problem may include the use of DERs, such as DG and DR, as well as the dispatch reprogramming of available generation resources. It is worth mentioning that the vulnerability assessment implemented in this paper has two stages. In the first stage, a plan of the best attack strategies for the interdiction vector is selected through a GA, following the logic depicted in Figure 1 and considering that the network does not count with DERs to protect against disruptive attacks. In the second stage,

the impact of DG and DR is evaluated to mitigate the risk of the network and improve its resiliency under these circumstances.

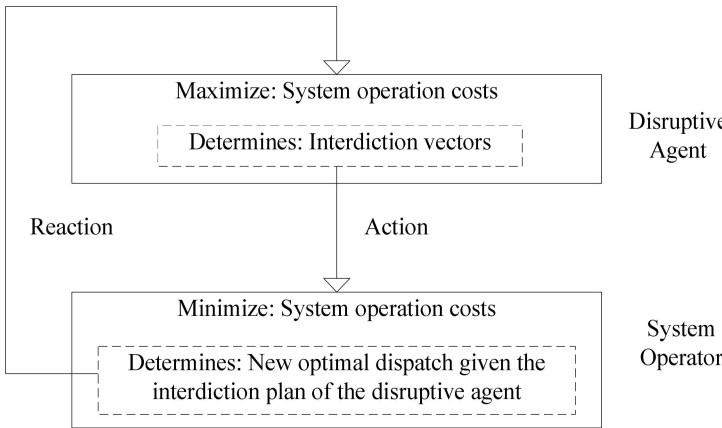

**Figure 1.** Bi-level programming diagram.

The attack plans of the disruptive agent are coded in a binary vector that indicates the states of the elements. This type of codification facilitates the implementation of a GA as solution strategy. In this case, the fitness of each individual within the GA corresponds to the system operation costs determined by the system operator. For this, an AC optimal power flow is executed.

Several metrics were established to quantify the resilience of the network and the case study scenarios that allow quantifying the effect of DR mechanisms and the location of DG in the network. DR is a mechanism that allows establishing bilateral agreements between the network operator and a set of loads for the voluntary disconnection of a percentage of their demand. According to [22], DR is defined as the amount of energy reduced in MWh with respect to the user's normal energy consumption, where changes in demanded energy are due to electricity price signals ($MWh) or incentive payments to reduce consumption when a contingency takes place.

In this case, an incentive payment scheme for voluntary load disconnection is used. Bilateral agreements are established between the network operator and a set of loads prior to the occurrence of a disruptive event and are applicable when the network operator loses the capacity to meet the demand of all loads. The cost assigned to each KWh with DR is normally set at a value greater than or equal to the cost of the generators but at a lower value than the mandatory disconnection of loads. As regards DG, this has become essential in the operation of networks that are isolated by the destruction of lines and generators, as well as in network topologies that require generation close to the loads to reduce power losses or increase the total generation capacity [23].

### 2.1. Normal Operative Conditions

In this stage, an optimal power flow under normal operating conditions is evaluated. There are also established the demands of each node, the marginal cost of the generators, the resources of the disruptive agent to perform attacks on the network, the costs of attacking lines and generators, the bilateral agreements for the voluntary disconnection of loads, the remuneration scheme for DR and the number and type of DG units that can be installed in the network to protect against possible attacks.

Figure 2 illustrates the percentage of load attended between periods t0 and t1 depending on the infrastructure that remains in operation after the execution of the disruptive event. The slope between t0 and t1 represents the loss of load due to an attack. Once this happens, the network operator executes an optimal dispatch to mitigate the effect of the attack, using the available network resources.

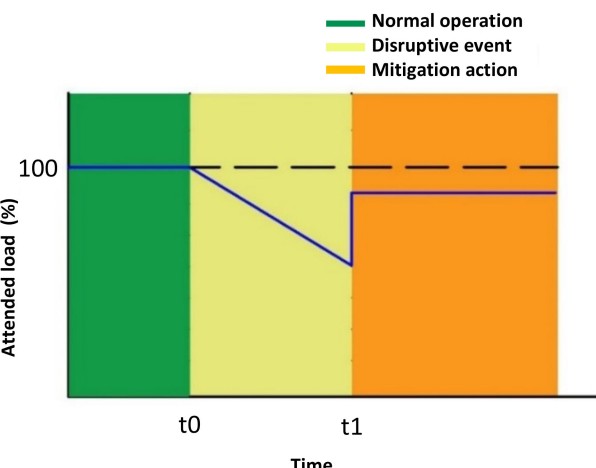

**Figure 2.** Resilience assessment phases.

The disruptive agent's attack strategy is selected through a vulnerability analysis that identifies the sensitivity of the network's operating cost to the failure of a line or generator. The vulnerability analysis developed in this paper acknowledges the following considerations:

- The resources that the network operator allocates for the protection of lines and generators in the system are known to the disruptive agent.
- The disruptive agent is aware of the bilateral agreements for voluntary load shedding.
- The attacks performed on the system are 100% effective.
- The network operator considers DR to be a mechanism for immediate mitigation of the attack plan and reduction of network operating costs.

The vulnerability analysis is applied to a case study with four scenarios that cover the strategies assumed by the network operator in the face of possible voluntary load disconnection agreements. In two of the four scenarios, the vulnerability of a network in which there is no DR in any of the loads is evaluated; in the remaining two scenarios, the vulnerability of a network in which there is a pact between some loads and the network operator to totally or partially reduce demand is evaluated. In the period between the execution of the attack (t0+) and the mitigation actions (t1−), it is possible to define a measure of resilience by minimizing the immediate effect of the attack through DR and the infrastructure that remains in operation within the network.

### 2.2. Scenarios for the Case Study

Resilience actions are strategies that are executed in the post-attack stage and involve the joint optimization of the use of the active infrastructure, the types of generators available and DR arrangements that were established prior to the development of the disruptive event. To evaluate the resilience that can be achieved in the possible scenarios derived from the use of DG and the DR mechanism, a case study with four scenarios is developed.

- Scenario 1: There is no agreement for voluntary load shedding. Under this condition, the most severe attack plan of the vulnerability analysis is executed without taking mitigation actions by the network operator.
- Scenario 2: There is a bilateral agreement between the network operator and some system loads to voluntarily disconnect a percentage of the total load. From this condition, the disruptive agent executes the most severe attack of the vulnerability analysis. In the post-attack stage, the network operator takes no action to decrease load shedding.
- Scenario 3: There is no agreement for voluntary load shedding. Under this condition, the most severe attack plan of the vulnerability analysis is executed. In the post-attack stage, the network operator optimizes the location and sizing of distributed generators to reduce load shedding.

- Scenario 4: There is a bilateral agreement between the network operator and some system loads to voluntarily disconnect a percentage of the total load. From the condition, the disruptive agent executes the most severe attack of the vulnerability analysis. In the post-attack stage, the network operator optimizes the location of distributed generators and reallocates demand response to reduce load shedding.

## 3. Mathematical Modeling and Solution Approach

### 3.1. Vulnerability Analysis

The vulnerability analysis of an electrical grid allows both a network operator and a disruptive agent to evaluate the lines and generators that can be attacked to generate the highest costs for mandatory load shedding and operation of conventional generators and different DERs in the grid. The vulnerability analysis starts from the normal operating conditions of the network, with the evaluation of the resources allocated by the network operator to protect each element of the network, the monitoring of voluntary disconnection contracts, the resources of the disruptor agent, and the costs for load shedding at each node. The dependence of an agent's decisions on those of its opponent makes it possible to describe the vulnerability analysis as a bi-level optimization model.

#### 3.1.1. Genetic Algorithm

GAs are metaheuristic techniques inspired by the Darwinian theory of evolution. These types of techniques have been successfully applied to solve bi-level programming problems [24–26]. Figure 3 depicts the flowchart of the implemented methodology, which includes the GA. In this case, a candidate solution or individual is represented by means of a binary vector that represents the lines and generators out of service.

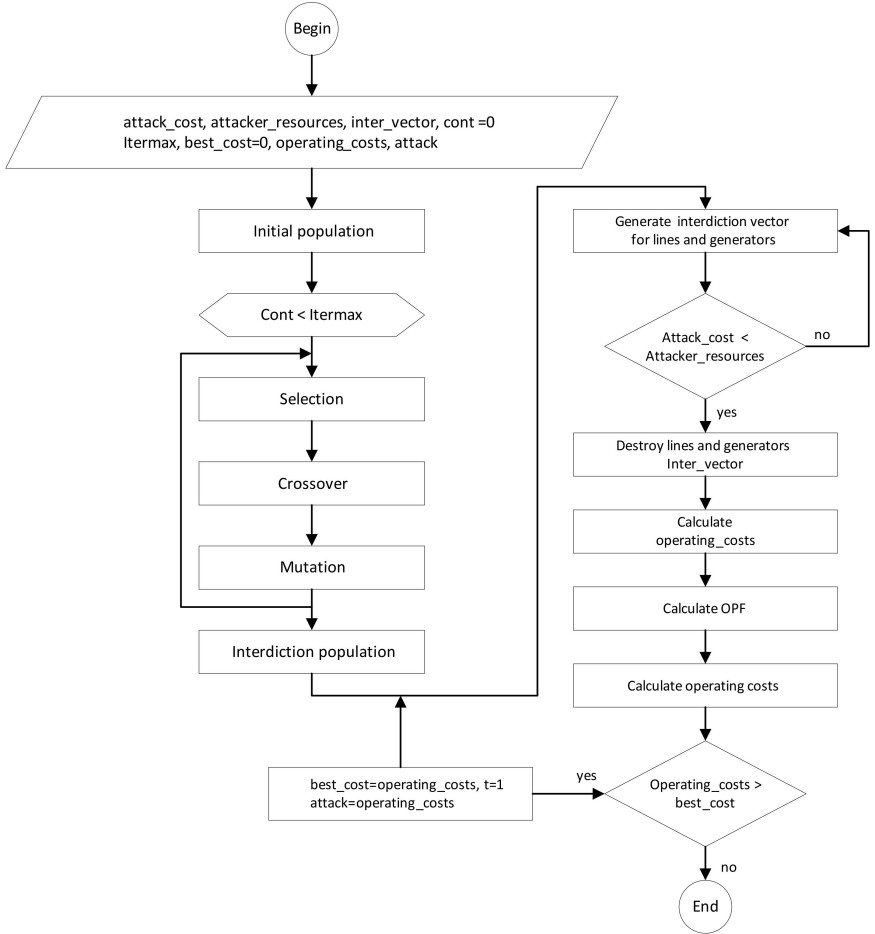

**Figure 3.** Flowchart of the proposed methodology.

The *initial population* consists of the random generation of interdiction vectors. Each interdiction vector represents an attack to the system and an individual or candidate solution within the GA. From the initial population, new candidate solutions are obtained by means of selection, recombination and mutation operators [27–29]. The selection mechanism guarantees that the best individuals are chosen to generate new candidate solutions. Such solutions are obtained by means of the recombination operator in which the selected parents interchange their genetic information. The mutation stage is in charge of adding diversity to the algorithm and eventually avoiding getting trapped in local optimal solutions. In every generation, the best solution candidates are preserved (those that maximize load shedding). The process stops after a given number of generations has elapsed [30].

### 3.1.2. Upper-Level Optimization Problem

The upper level optimization problem is given by Equations (1)–(4). The objective function illustrated in Equation (1) is to maximize the operating cost of the network after an attack. In this case, $C_g$ is the cost of the power delivered by generator g; $P_g$ is the power delivered by g; $C_{RD_n}$ is the cost of the dispatchable load at node $n$; $P_{RD_n}$ is the demand response at the node $n$; $P_{D_m}$ is the load shedding at node $m$; and $C_{D_m}$ is the cost of load shedding at node $m$. The disruptive agent strategy is modeled through an interdiction vector for lines and generators. In Equation (2) and Equation (3), the interdiction vector for lines and generators, respectively, is defined as a vector of binary variables, where 1 represents the elements under attack. In this case, $\delta_L(l)$ and $\delta_G(g)$ are the interdiction vectors for the set of lines and generators, respectively. Constraint Equation (4) describes the destructive resources of the attacking agent, where $M_l$ is the cost of attacking a line, while $M_g$ is the cost of attacking a generator. L is the set of lines; G is the set of generators; $M$ represents the total resources of the attacker; $N$ is the set of buses; and $NRD$ is the set of buses with the demand response.

$$Max \quad Z = \sum_g C_g\, P_g + \sum_n C_{RDn}P_{RDn} + \sum_m P_{Dm}C_{Dm}$$
$$\forall\, g \in G\, ,\ \forall\, n \in NRD,\ m \in N \tag{1}$$

$$\delta_L(l) \in \{0,1\}; \forall\, l \in L \tag{2}$$

$$\delta_G(g) \in \{0,1\}; \forall\, g \in G \tag{3}$$

$$\sum_l M_l\delta_L(l) + \sum_g M_g\delta_G(g) \leq M;\ \forall\, l \in L\, , \forall\, g \in G \tag{4}$$

### 3.1.3. Lower-Level Optimization Problem

The lower-level optimization problem defines the response of the system operator through the calculation of an optimal AC power dispatch. Equation (5) presents the objective function of the network operator. It considers the cost of operation of the available generators, the cost of voluntary load disconnection through the DR mechanism and the cost of mandatory load disconnection. Note that in this case, the problem is of minimization. Constraints Equation (6) to (10) define the physical characteristics of the network related to the limits of active, reactive, and apparent power in generators, loads and lines, respectively. In this case, $Q_g$ is the reactive power delivered by generator g; $P_d$ and $Q_d$ are the active and reactive power demand, respectively; and $S_l^{Br}$ is the apparent power flow in line $l$. Equations (11) and (12) describe the active power flow balance at each node, incorporating the disconnected active power limit (voluntary and compulsory) at loads, as well as the interdiction vector described in Equation (2), and Equation (3). In this case, $P_{RD_n}$ is the demand response at node n; $P_{D_m}$ is the load shedding at node m; and $Wn$ is the power scheduled for generator n. In Equations (13) and (14), the active and reactive power transmitted through the lines are represented. Equations (15) and (16) indicate the active and reactive power balance, respectively; $\Psi_G{}^n$ is the set of generators connected to node n; and $\Psi_D{}^n$ is the set of demands connected to node and $\Psi_L{}^n$ is set of lines connected

to node n. The lower-level optimization model establishes the optimal power flow in the network after the attack and the system operation cost represented by variable Z.

$$Min \quad Z = \sum_g C_g P_g + \sum_n C_{RDn} P_{RDn} + \sum_m P_{Dm} C_{Dm} \tag{5}$$

$$\forall\, g \in G\,,\ \forall\, n \in NRD,\ m \in N$$

$$P_g^{min} < P_g < P_g^{max} \tag{6}$$

$$Q_g^{min} < Q_g < Q_g^{max} \tag{7}$$

$$0 < P_d < P_d^{max} \tag{8}$$

$$0 < Q_d < Q_d^{max} \tag{9}$$

$$S_l^{Br\ min} < S_l^{Br} < S_l^{Br\ max} \tag{10}$$

$$P_d + P_{RDn} + P_{Dm} = P_d^{max} \tag{11}$$

$$0 < P_{RDn} \le P_d^{max} W_n RD_n \tag{12}$$

$$P_{sr} = |V_s|^2 g_{sr} - |V_s||V_r|\cos g_{sr}\cos(\delta_s - \delta_r) - |V_s||V_r| b_{sr} sen(\delta_s - \delta_r) \tag{13}$$

$$Q_{sr} = |V_s|^2 b_{sr} + \cos b_{sr}|V_s||V_r|\cos(\delta_s - \delta_r) - |V_s||V_r| b_{sr}\, g_{sr} sen(\delta_s - \delta_r) \tag{14}$$

$$\sum_{\forall g \in \Psi_G{}^n} (1 - \delta_G(g))P_g - \sum_{\forall d \in \Psi_D{}^n} P_d - \sum_{\forall s \in \Psi_L{}^n} \delta_l^{Br} P_{sr} = 0 \tag{15}$$

$$\sum_{\forall g \in \Psi_G{}^n} (1 - \delta_G(g))Q_g - \sum_{\forall d \in \Psi_D{}^n} Q_d - \sum_{\forall s \in \Psi_L{}^n} \delta_l^{Br} Q_{sr} = 0 \tag{16}$$

### 3.2. Allocation of Costs in Loads

The disruptive agent attack plan is represented as an interdiction vector for lines and generators in binary coding, where an attacked element is expressed by Equation (1). The costs assigned to the generators are defined by a first order polynomial model; the load shedding and DR costs at each of the nodes are determined by a piecewise linear model. Unlike the polynomial model, the piecewise linear model allows characterizing the load cost as a function of discrete conditions [31]. Figure 4 and Equations (17)–(19) represent the scheme for cost allocation in loads where $c1$ *y* $c2$ are the uncertainty costs, $x1$ and $x2$ are the power demanded, and $m1$ *y* $m2$ are the results of the operation of costs and minimum and maximum power at the loads.

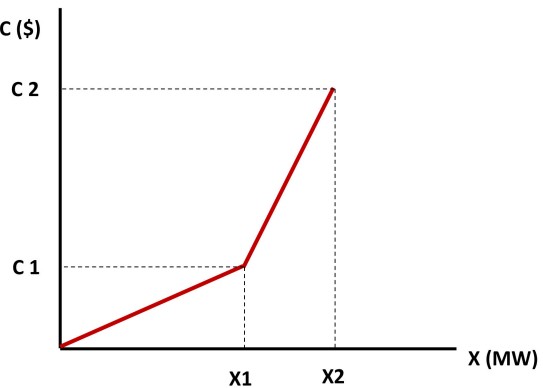

**Figure 4.** Variable cost allocation of loads.

$$(x) = \begin{cases} 0, & x = 0 \\ m1x, & 0 < x \le x1 \\ m2\,(x - x1) + c1, & x1 < x \le x2 \end{cases} \tag{17}$$

$$m1 = \frac{c1}{x1} \qquad m2 = \frac{c2 - c1}{x2 - x1} \tag{18}$$

$$x = P_d^{max} - Pd \qquad x1 = P_d^{max} - RD \qquad x2 = P_d^{max} \tag{19}$$

### 3.3. Resilience of an Electrical Power System

The resilience of an electrical power system involves preparing for, responding to and mitigating attacks that affect an electrical grid. A qualitative assessment of the resilience of a power grid requires evaluating the effectiveness of the measures taken and the comparison of different response strategies by the network operator. Several metrics are used to assess and calculate the resilience of an electric power system [32]. Some are designed to improve engineering resilience at the asset level, which are usually physical in nature and do not need human intervention for their application; others are operational resilience metrics, which focus on system-level performance and characteristics intended to mitigate the risk of failure and support service recovery [33]. Another example of operational metrics is that, due to interruptions in operation and availability of power served per total amount of power demand [31], the metrics should reflect system performance and be useful for decision making [22,34,35].

In this paper, three metrics are proposed to evaluate the resilience of power grids, which are based on the total load served and the operating cost of the grid, respectively. Equation (20) presents the resilience metric based on the total load served, which allows measuring the effectiveness of mitigation actions to reduce the effect of the disruptive attack.

$$\mu1 = \frac{Served\ Load}{Total\ Load} \tag{20}$$

In the $\mu1$ metric, a value close to 1 represents the ability of the system to adequately manage the optimal power flow to meet the demand. On the other hand, a value close to 0 defines the worst case scenario for the network in which the power supply is minimal.

Equation (21) presents the $\mu2$ metric, which measures the efficiency of the network operator's mitigation actions to minimize the cost of operating the network after an attack by evaluating the portion of the total network operating cost that corresponds to load shedding.

$$\mu2 = 1 - \frac{Load\ Shedding\ Cost}{Operation\ Cost} \tag{21}$$

In the $\mu2$ metric, a value close to 1 shows that the power grid has mechanisms to minimize mandatory load shedding as the worst case scenario for the network operator and the loads, whereas a value close to 0 coincides with a scenario in which the cost of load shedding is considerably higher than the operating costs of generators and the cost of voluntary load shedding.

In Equation (22), the metric $\mu$ is proposed as the measure of resilience of the topology and characteristics of the power grid to the most severe disruptive event that an attacker can develop. In this metric, the ability of mitigation actions to decrease load shedding and the cost of operating the grid after an attack is jointly evaluated.

$$\mu = \frac{\mu1 + \mu2}{2} \tag{22}$$

Values of $\mu$ equal to 1 and 0 quantify, respectively, a fully resilient and a zero resilient network. Although the mandatory disconnection implies increased costs, the discrimination of load shedding costs introduces a relationship that is not necessarily directly proportional. Table 1 allows the network operator to generate a quantitative and qualitative assessment of the resilience of an electrical grid.

**Table 1.** Qualification of resiliency.

| $\mu$ | Resiliency Degree |
|---|---|
| $\mu = 0$ | None |
| $0 < \mu \leq 0.25$ | Deficient |
| $0.25 < \mu \leq 0.5$ | Poor |
| $0.5 < \mu \leq 0.75$ | Regular |
| $0.75 < \mu < 1$ | Good |
| $\mu = 1$ | Excellent |

*3.4. Strategies for Maximizing Network Resilience Following a Disruptive Event*

This section describes the model that jointly optimizes the location of DG and the DR mechanism as mitigation actions against disruptive events. Based on the attack that the disruptive agent selected from the vulnerability analysis, the optimal placement and sizing of DG, as well as DR, are evaluated. The objective function of the optimization model for DG location is described in Equation (23); in comparison to Equation (5), the objective function for this model incorporates the cost of DG. The constraints that are associated with active and reactive power limits on generators, loads and lines are taken from Equation (6)–(12). Equations (24) and (25) are the active and reactive power balance at each node, while Equations (26) and (27) define the maximum power generated by the DG. In this case, $C_g$ and $P_g$ are the cost and power delivered by generator g, respectively. $C_{gd}$ and $P_gd$ are the cost of demand and power demanded, respectively. $C_{RDn}$ and $P_{RDn}$ are the cost and amount of DR, respectively. $P_{Dm}$ and $C_{Dm}$ are the demand at node $m$ and its costs, respectively. $P_g$ is the active power supplied by generator $g$, $P_{gd}$ is the active power supplied by DG, and $P_d$ is the active power that is demanded. Finally, Equation (28) establishes the maximum number of DG units that can be used within the optimization process. In this case, $Q_g$ and $Q_d$ are the generation and demand of reactive power, and $V_n$ is the voltage magnitude at bus n.

$$Min \quad Z = \sum_g C_g \, P_g + \sum_{gd} C_{gd} \, P_{gd} + \sum_n C_{RDn} P_{RDn} + \sum_m P_{Dm} C_{Dm} \tag{23}$$

$$\forall \, g \in G \, , \, \forall \, n \in NRD, \, m \in N$$

$$\sum_{\forall g \in \Psi_G{}^n} P_g + \sum_{\forall gd \in \Psi_{GD}{}^n} P_gd - \sum_{\forall d \in \Psi_D{}^n} P_d = V_n \sum_{j \in \Omega N} V_j Y_{ij} cos(\delta j - \delta i + \theta ij) \tag{24}$$

$$\sum_{\forall g \in \Psi_G{}^n} Q_g - \sum_{\forall d \in \Psi_D{}^n} Q_d = V_n \sum_{j \in \Omega N} V_j Y_{ij} sin\left(-\delta j + \delta i - \theta ij\right) \tag{25}$$

$$0 \leq P_{gd} \leq x_{gd} P_{gd}^{max} \tag{26}$$

$$x_{GD}(gd) \in \{0, 1\}; \forall \, gd \, \in \, GD \tag{27}$$

$$\sum\nolimits_{gd} x_{gd} \leq N_{gd}^{max} \tag{28}$$

**4. Tests and Results**

To show the applicability and effectiveness of the proposed approach, several tests were carried out with a didactic 5-bus power system and the IEEE RTS 24-bus power system.

*4.1. Results with a 5-Bus Power System*

The proposed vulnerability analysis was initially applied to a power system composed of five buses and five generators for which data can be consulted in [36]. This power system is illustrated in Figure 5.

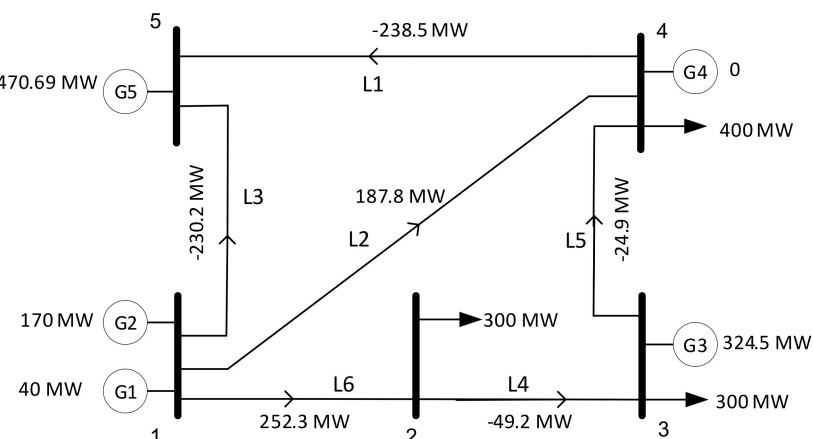

**Figure 5.** Illustration of power flows in the test system.

### 4.1.1. Normal Operation and Disruptive Event

The normal operating characteristic of the system is calculated with an optimal power flow that minimizes the cost of supplying the load. The data of the system and results of the power flow are presented in Tables 2 and 3. Under normal operating conditions, the system has a total capacity of 1530 MW and a total demand of 1000 MW. In this case, line 6 and generators 1 and 6 are operating at their maximum limits.

**Table 2.** Bus data.

| Bus | Type | P Load (MW) | Q Load (MVAR) | Voltage (p. u) |
|---|---|---|---|---|
| 1 | PV | 0 | 0 | 1.07 |
| 2 | PQ | 300 | 98.61 | 1.08 |
| 3 | PV | 300 | 98.61 | 1.09 |
| 4 | Slack | 400 | 131.47 | 1.06 |
| 5 | PV | 0 | 0 | 1.06 |

**Table 3.** Power dispatch of generators.

| Generator | Pg (MW) | Qg (MVAR) | Cost (USD/MWh) |
|---|---|---|---|
| G1 | 40 | 30 | 14 |
| G2 | 170 | 127.5 | 15 |
| G3 | 324.5 | 390 | 30 |
| G4 | 0 | −10.8 | 40 |
| G5 | 470.69 | −165 | 10 |

The effectiveness of attacks performed to maximize load shedding and increase the cost of network operation depends on the number of resources of the disruptive agent, the cost associated with attacking a set of lines or generators, and the resilience of the network after the attack. Network resilience is quantified by the ability to meet the scheduled load and minimize the increase in the cost of operating the network with the infrastructure remaining in operation after the attack. The costs related with the network operator and the disruptive agent are presented in Table 4. In this case, the charging mechanism for mandatory load disconnection (load shedding) allows the user to receive compensation for the cost of each MWh that is not being delivered and establishes a hierarchy of loads based on the cost of load shedding.

**Table 4.** System resources.

| Resource | Costs of Attacking Elements, DR, and Load Shedding |
|---|---|
| Cost of attacking a line (USD/line) | 50 |
| Cost of attacking a generator (USD/generator) | 100 |
| Total resources of the attacker (USD) | 300 |
| Load shedding costs buses 2, 3, 4 (USD/MWh) | 100, 100, 400 |
| DR at buses 2, 3, 4 (%) | 0, 50, 25 |
| Cost of DR at buses 2, 3, 4 (USD/MWh) | 0, 50, 50 |

From the allocation of the load shedding cost and the generators' operating cost, a proportional relationship is established between the network operating cost and the total load shedding cost after the disruptive event. Table 5 presents the top ten attack plans of the operation of the network after the execution of the best attack plan of the disruptor agent. From these results, it is evident that the best attack plans are those in which load shedding occurs at node 4 since this is a critical load. A critical load is considered to be those infrastructures that are associated with the basic needs of human life, which include hospitals, public lighting, water utilities, telecommunications and others [37,38].

**Table 5.** Best attack plans.

| Attack | Attacked Lines | Attacked Generators | Operation Cost $\times 10^5$ (USD) | % of Load Served |
|---|---|---|---|---|
| 1 | L1, L2, L5, L6 | G4 | 1.8365 | 52 |
| 2 | L2, L3, L5, L6 | G4 | 1.7485 | 60 |
| 3 | L2, L4, L5, L6 | G4 | 1.7203 | 60 |
| 4 | L2, L5, L6 | G4 | 1.7013 | 60 |
| 5 | L1, L2, L6 | G3 | 1.4800 | 20 |
| 6 | L1, L2, L3, L6 | G3 | 1.4800 | 20 |
| 7 | L1, L2, L4, L6 | G3 | 1.4800 | 20 |
| 8 | L1, L2, L5, L6 | G3 | 1.4800 | 20 |
| 9 | L1, L2 | G3, G4 | 1.3287 | 22.34 |
| 10 | L2, L3, L4, L6 | G3 | 1.3023 | 40.88 |

Figure 6 and Table 6 summarize the most relevant characteristics of the network operation after the execution of the best attack plan of the disruptor agent. This attack generates three islands that prevent the connection between the operating generators and the demands. The operating cost after the most severe attack is USD $1.8365 \times 10^5$, where load shedding represents 93.5% of the total cost and only 20% of the load is served.

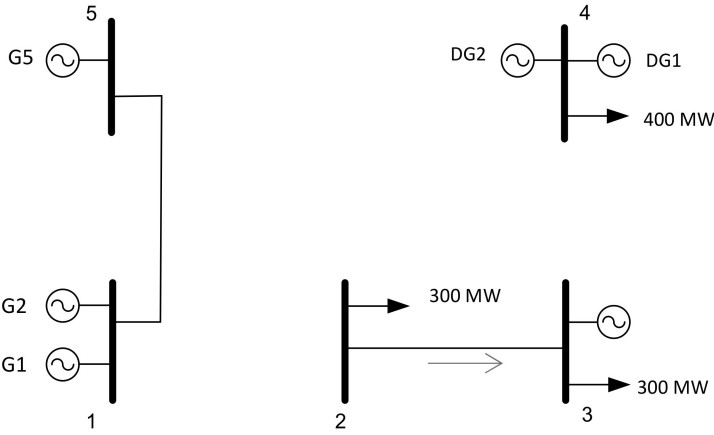

**Figure 6.** Topology of the system after the worst attack.

**Table 6.** Description of served loads and load shedding costs.

| Bus | Generation (MW) | Generation Cost (USD) | Served Load (MW) | Served Load (%) | Cost of Load Shedding (USD) |
|---|---|---|---|---|---|
| 1 | 0 | 0 | - | - | - |
| 2 | - | - | 220 | 73.33 | 8000 |
| 3 | 520 | 15,600 | 300 | 100 | 0 |
| 4 | 0 | 0 | 0 | 0 | 160,000 |
| 5 | 0 | 0 | - | - | - |

### 4.1.2. Disruptive Event with DR

DR can be considered as an immediate reaction mechanism to network contingencies. Therefore, this section performs a vulnerability analysis in which there are bilateral agreements between the network operator and the disruptor agent for voluntary load disconnection in case of a disruptive event. This strategy can decrease the load shedding and the network operation cost even when the disruptor agent is aware of such agreements. In this case, it is supposed that through the DR mechanism, a bilateral agreement is allowed to voluntarily disconnect 50% of the load from node 3 and 25% of the load connected to node 4 when a disruptive event occurs in the network. Tables 7 and 8 describe the results of the vulnerability analysis with the costs described in Table 4. The impact of the DR mechanism on the resilience of the network is represented by the decrease in network operating cost and load shedding, compared to the results of the attack with higher severity presented in Section 4.1.2.

**Table 7.** Description of the attack plan.

| Attacked Elements | Operation Cost (USD) | Served Load (MW) |
|---|---|---|
| L1, L2, L5, L6, G4 | 14,464 | 700 |

**Table 8.** Power dispatched and served load.

| Node | Power Supplied (MW) | DR (MW) | Load Shedding (MW) | Cost (USD) | Served Load (%) |
|---|---|---|---|---|---|
| 2 | 300 | 0 | 0 | 0 | 100 |
| 3 | 220 | 80 | 0 | 4000 | 100 |
| 4 | 0 | 100 | 300 | 125,000 | 25 |

### 4.1.3. DG Allocation to Increase Resiliency without DR

The location of DG in a network reduces load shedding and network operating costs. They are an alternative for the post-disruptive event stage since, compared to the recovery of the lost infrastructure, they are more efficient in terms of costs and installation times. This section identifies the optimal location and sizing when the most severe disruptive attack described in Section 4.1.1 occurs. The types of DG units selected are presented in Table 9. Table 10 shows how the installation of DG units at node 4 improves the grid resilience by decreasing the grid operating cost and increasing the load served, contrasted to the results of Section 3.2. Table 11 describes the network operation characteristics after the disruptive event and DG location. In comparison with Table 8 serving 100% of the load at node 4, DG represents for the network a reduction of USD 142,000 at node 4 alone.

**Table 9.** Types of DG.

| Type | Pmax (MW) | Cost (USD/MW) |
|---|---|---|
| 1 | 100 | 45 |
| 2 | 300 | 45 |

**Table 10.** Operation after locating DG.

| Location DG-Type | Power Generated (MW) | Operation Cost (USD) | Served Load (MW) | Load Supplied (%) |
|---|---|---|---|---|
| 4(1), 4(2) | 100, 300 | 41,648 | 920 | 92 |

**Table 11.** Description of loads at buses 2, 3 and 4.

| Bus | Active Power (MW) | Load Shedding (MW) | Cost (USD) | Load Supplied (%) |
|---|---|---|---|---|
| 2 | 300 | 0 | 0 | 100 |
| 3 | 220 | 80 | 8000 | 73.33 |
| 4 | 400 | 0 | 0 | 100 |

4.1.4. Location and Sizing of DG along with DR to Improve Resiliency

This section evaluates the placement and sizing of DG in conjunction with the DR mechanism to increase the operational resilience of the grid when the disruptive event described in Section 4.1.2 occurs. Table 12 summarizes the location and optimal sizing of DERs in the network.

**Table 12.** Sizing of distributed energy resources.

| Location DG-Type | DR Location | DR | Operation Cost (USD) | Load Supplied (MW) |
|---|---|---|---|---|
| 4-(1), 4-(2) | 3 | 80 | 37,645 | 1000 |

The results presented show that the system operator minimizes the damage caused to the grid when there is an agreement to voluntarily disconnect a percentage of load 3 and the two types of distributed generators available at node 4 are located. The demand response mechanism contributes to serving 8% of the total load on the grid and decreases the operating costs at node 3 from USD 8000 to USD 4000, while the distributed generators at node 4 serve 40% of the total load on the grid and decrease the operating costs at node 4 from USD 160,000 to USD 18,000.

4.1.5. Quantification of Resilience in Terms of Operating Cost and Percentage of the Total Load

This section presents a comparison of the grid operation, according to the management of DG and DR developed in Sections 3.1 and 3.2. Table 13 quantifies resilience in terms of operating cost and percentage of total load served, and Table 14 describes quantitatively and qualitatively the level of resilience achieved with each proposed scenario, which is shown in Figure 7. This figure shows the level of load served in the development of the disruptive event (t0–t1) and in the stage after the event (t1–t2) with respect to normal operating conditions.

**Table 13.** Scenario results according to costs served load.

| | Served Load (MW) | Operation Cost (USD) | Load Shedding Cost (USD) |
|---|---|---|---|
| Scenario 1 | 520 | 183,650 | 168,000 |
| Scenario 2 | 700 | 144,645 | 120,000 |
| Scenario 3 | 920 | 41,648 | 8000 |
| Scenario 4 | 1000 | 37,645 | 0 |

**Table 14.** Resilience metrics.

|  | $\mu 1$ | $\mu 2$ | $\mu$ | Resilience |
| --- | --- | --- | --- | --- |
| Scenario 1 | 0.52 | 0.0852 | 0.3026 | Poor |
| Scenario 2 | 0.70 | 0.1703 | 0.4351 | Poor |
| Scenario 3 | 0.92 | 0.8079 | 0.8639 | Good |
| Scenario 4 | 1 | 1 | 1 | Excellent |

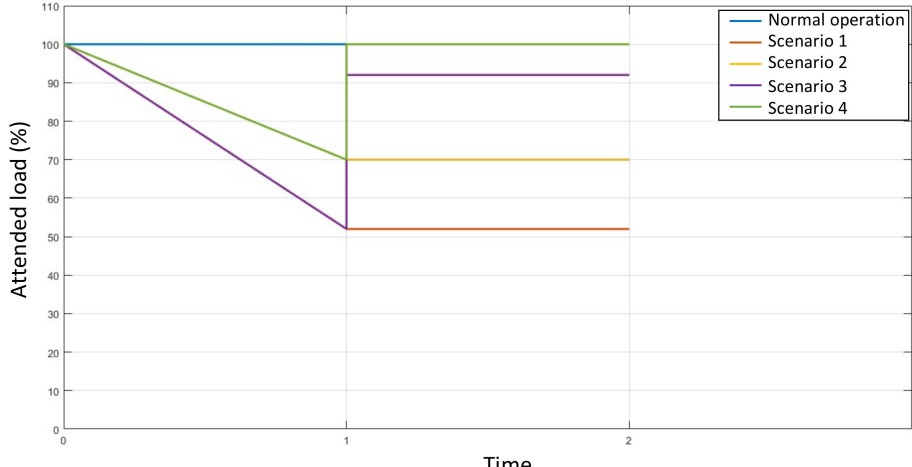

**Figure 7.** Served load in percentage.

It is important to remember that in Scenario 1, the grid is attacked, and no mechanism is used to respond to this attack. In Scenario 2, there is a bilateral agreement between the network operator and some loads to voluntarily disconnect a percentage of the load; in the post-attack stage, the network operator does not take actions to reduce load shedding. In Scenario 3, there is no agreement for voluntary load shedding; under this condition, the attack plan is executed. In the post-attack stage, the network operator optimizes the location and sizing of distributed generators to reduce load shedding. In Scenario 4, there is a bilateral agreement between the network operator and some system loads to voluntarily shed a percentage of the total load. In the post-attack stage, the network operator optimizes the location of distributed generators and reallocates the demand response to reduce load shedding.

### 4.2. Results with the IEEE RTS-24 Bus Power System

In this section, the proposed methodology is applied to a modified version of the IEEE RTS-24 bus test system. Figure 8 illustrates the notation assigned to each of the network elements as well as the operating costs of the generators in USD/MWh. Note that the nodes with DR are indicated in blue circles. It is supposed that the cost of attacking a line or generator is the same, indicated in Table 4, while the cost of DR is 50 USD/MWh; also, the disruptive agent has a budget of USD 800 to design the attack plans. The cost of load shedding is 100 USD/MWh at all nodes, with the exception of buses 2, 9, 15, 16, 19 and 20 where it is 300 USD/MWh. A maximum of 6 DG units might be allocated with capacity of 40 MW each. The first three DG units (type 1) have a cost of 40 USD/MWh and the others (type 2) a cost of 45 USD/MWh. These can be allocated at any of the load buses indicated in Figure 8.

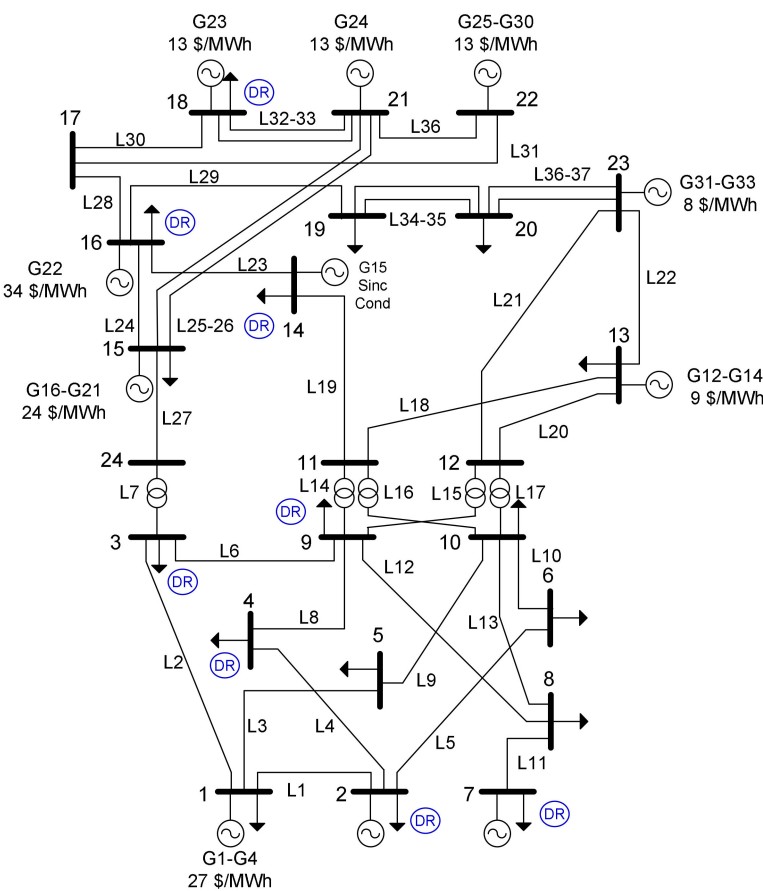

**Figure 8.** Modified version of the IEEE RTS-24 bus test system.

Under normal operating conditions, the total load served is 2850 MW and the total operating cost is USD 38,284. The resources allocated to the disruptive agent allow it to execute attacks plans that results in different levels of load shedding. Four cases are considered as indicated in Table 15. In Scenario 1, the percentage of served load is only 38% and the operative cost is USD 336,464, due to load shedding. For Scenario 2, the system operator establishes bilateral agreements for voluntary load disconnection (DR) and reaches a percentage of served load of 54%. In this case, DR allows serving 15% of the total demand and has a cost of USD 13,325. In Scenario 3, DG is used to mitigate the effect of the disruptive event on the grid. Locating the 6 DG units at the most affected buses allows meeting a percentage of 8% of the total load supplied. In Scenario 4, using both DG and DR together, the highest level of resilience is achieved, reducing the total cost of the system and the amount of load shedding in relation to the previous ones; 61% of the total load of the network is served, and together, these mechanisms represent 23% of the load served. Tables 15 and 16 summarize the vulnerability and resilience results for the four cases. Table 17 presents the resilience metrics in each of the cases when presented with a disruptive attack.

**Table 15.** Attacked elements and use of DERs.

|  | Attacked Lines | Attacked Generators | DG (Bus-Type) | DR (Bus) |
|---|---|---|---|---|
| **Scenario 1** | 1, 7, 10, 15, 17, 18, 19, 25, 26, 28, 36, 37 | G21, G22 | - | - |
| **Scenario 2** | 1, 2, 3, 8, 10, 11, 18, 20, 21, 23, 27, 29, 36 | G23 | - | 6, 8, 9, 11, 14, 20 |
| **Scenario 3** | 1, 7, 10, 15, 17, 18, 19, 25, 26, 28, 36, 37 | G21, G22 | 6-1, 9-1, 12-1, 14-2, 19-2, 24-2 | - |
| **Scenario 4** | 1, 2, 3, 8, 10, 11, 18, 20, 21, 23, 27, 29, 36 | G23 | 6-1, 9-1, 12-1, 14-2, 19-2, 24-2 | 6, 8, 9, 11, 14, 20 |

**Table 16.** Summary of costs.

|            | Load Served (MW) | Load Served (%) | Load Shedding Cost (USD) |
| ---------- | ---------------- | --------------- | ------------------------ |
| Scenario 1 | 1094.5           | 38.40           | 336,464                  |
| Scenario 2 | 1527.2           | 53.58           | 218,736                  |
| Scenario 3 | 1318.0           | 46.24           | 255,394                  |
| Scenario 4 | 1750.7           | 61.42           | 184,442                  |

**Table 17.** Resiliency metrics.

|            | $\mu$  | $\mu2$   | $\mu$   | Resiliency |
| ---------- | ------ | -------- | ------- | ---------- |
| Scenario 1 | 0.3840 | 0.052211 | 0.21812 | Deficient  |
| Scenario 2 | 0.5359 | 0.131355 | 0.33360 | Poor       |
| Scenario 3 | 0.4625 | 0.078019 | 0.27023 | Poor       |
| Scenario 4 | 0.6142 | 0.147741 | 0.38095 | Poor       |

## 5. Discussion of Results

The proposed GA showed to be effective at solving the EGIP modeled in this work. For the 5-bus test system, the average execution time was 2 min and 42 s, while for the IEEE RTS 24-bus test system, it was 29 min and 50 s. All tests were carried out on a laptop under a Windows 10 operating system with an i7 Pentium Core processor and 8 GHz of RAM memory.

In the 5-bus test system, it was observed a poor resilience in Scenario 1 where no DERs were considered to mitigate the damage caused to the network by the malicious attack. In Scenario 2, considering only DR, an increment in resilience was observed; nonetheless, it still remained below the threshold of 0.5 to be at least regular as given by the metric proposed in this work (see Table 1). This value increased importantly when DG was introduced in the system (Scenario 3) to serve the load isolated at bus 4, decreasing significantly the total load shedding. Finally, an excellent resilience was achieved in Scenario 4 when both resources, namely DG and RD, were combined. In this last Scenario, there was no load shedding, despite the attacked elements. This highlights the importance of an optimal assignment of DERs performed by the system operator.

The tests carried out with the IEEE RTS-24 bus test system showed that the resiliency could not be highly improved, despite having DERs available. This is basically due to two facts. On one hand, it was assigned a high budget for the attacker, which was able to destroy a significant portion of the system (an average of twelve lines and at least one generator). On the other hand, the total power provided by the DG units was limited to 8% of the total demand, and the bilateral agreements of voluntarily load disconnection (DR) was also limited to a reduced set of loads that added up 15% of the total demand. In this case, despite the optimal allocation of DERs performed by the system operator, the resiliency index remained below 0.5. Nonetheless, an important reduction in load shedding costs was achieved. According to the data presented in Table 16, the load served increased from 38.4% to 61.42%, which represents important savings in load shedding costs.

## 6. Conclusions

This paper addresses the electric grid interdiction problem in which a malicious agent aims at causing maximum damage to the network subject to a limited budget and the reaction of the system operator that may resort to DERs in order to mitigate the impacts on the network. This attacker–defender dynamic is modeled within a bi-level programming framework and solved through a GA. The vulnerability analysis presented in this paper allows identifying the best strategies of the disruptor agent to attack the lines and generators of a network, and the response of the system operator to minimize load shedding and operating costs based on the location and sizing of distributed generators in conjunction with the demand response mechanism in order to increase the operational resilience of the network when the disruptive event occurs. In addition, the interaction between the

disruptive agent and the network operator is illustrated in terms of the economic and physical constraints of the grid.

The metrics proposed by the authors allow measuring the actions taken by the system operator regarding load shedding and distributed energy resources as a demand response, the distributed generation to be used for system resilience, and the costs that this represents for the grid and the user. The results presented show that the system operator minimizes the damages caused to the grid as well as the operating costs when there is an agreement to voluntarily disconnect a percentage of the load and distributed generators are located at certain nodes in order to meet the demand.

Future work may include other power flow models, and new metrics for resiliency as well as the use of novel techniques to approach the bi-level programming framework that models the attacker–defender dynamic. A management system that allows the system operator to prioritize critical loads in the event of a deliberate attack may also be implemented. Finally, a future challenge is the implementation of other game-theoretic approaches, such as tri-level optimization, that also captures the response of the system planner.

**Author Contributions:** Conceptualization, D.J.M.P., E.R.T. and J.M.L.-L.; Data curation, D.J.M.P., E.R.T. and J.M.L.-L.; Formal analysis, D.J.M.P., E.R.T. and J.M.L.-L.; Funding acquisition, D.J.M.P., E.R.T. and J.M.L.-L.; Investigation, D.J.M.P., E.R.T. and J.M.L.-L.; Methodology, D.J.M.P., E.R.T. and J.M.L.-L.; Project administration, D.J.M.P., E.R.T. and J.M.L.-L.; Resources, D.J.M.P., E.R.T. and J.M.L.-L.; Software, D.J.M.P., E.R.T. and J.M.L.-L.; Supervision, D.J.M.P., E.R.T. and J.M.L.-L.; Validation, D.J.M.P., E.R.T. and J.M.L.-L.; Writing—original draft, D.J.M.P., E.R.T. and J.M.L.-L.; Writing—review and editing, D.J.M.P., E.R.T. and J.M.L.-L.; All authors have read and agreed to the published version of the manuscript.

**Funding:** This research was funded by Centro de Investigación y Desarrollo Científico de la Universidad Distrital Francisco José de Caldas under code 2-5-604-19 associated with the project "Gestión de Recursos Energéticos Distribuidos (DER) y monitoreo de señales sísmicas en Situación de Desastres", and Colombian Scientific Program within the framework of the call Ecosistema Científico (Contract No. FP44842-218-2018).

**Acknowledgments:** The authors thank the C.I.D.C (Centro de Investigación y Desarrollo Científico) and Doctorate in Engineering of Universidad Distrital Francisco José de Caldas for the support on the development of this work. As well as the GCEM research group of the Universidad Distrital Francisco José de Caldas and the GIMEL research group of the Universidad de Antioquia where the research internship on this project is carried out. Finally, the authors would like to acknowledge the support from the Colombian Scientific Program within the framework of the call Ecosistema Científico (Contract No. FP44842-218-2018).

**Conflicts of Interest:** The authors declare no conflict of interest.

## Acronyms and Abbreviations

| AC | Alternating Current |
| DC | Direct Current |
| DERs | Distributed Energy Resources |
| DG | Distributed Generation |
| DR | Demand Response |
| EPS | Electric Power Systems |
| GA | Genetic Algorithm |
| PSO | Particle Swarm Optimization |

## Nomenclature

*Indices and sets*

| $L$ | Set of lines |
| $G$ | Set of generators |
| $M$ | Total resources of the attacker |
| $N$ | Set of buses |

| | |
|---|---|
| $NRD$ | Set of buses with demand response |
| $\Psi_G{}^n$ | Set of generators connected to node $n$ |
| $\Psi_D{}^n$ | Set of demands connected to node $n$ |
| $\Psi_L{}^n$ | Set of lines connected to node $n$ |
| *Parameters and constants* | |
| $C_g$ | Cost of the power delivered by generator $g$ |
| $C_{RDn}$ | Cost of the dispatchable load at node $n$ |
| $C_{Dm}$ | Cost of load shedding at node $m$ |
| $M_l$ | Cost of attacking a line |
| $M_g$ | Cost of attacking a generator |
| $c1, c2$ | Uncertainty costs |
| $C_{gd}$ | Cost of demand |
| $P_g d$ | Power demanded |
| *Variables* | |
| $P_g$ | Power delivered |
| $P_{RDn}$ | Power demand response at node n |
| $P_{Dm}$ | Power Load shedding at node m |
| $Q_g$ | Reactive power delivered by generator g |
| $\delta_L(l)$ | Interdiction vector for the set of lines |
| $\delta_G(g)$ | Interdiction vectors for the set of generators |
| $P_d$ | Active power demand |
| $Q_d$ | Reactive power demand |
| $S_l{}^{Br}$ | Apparent power flow in line l |
| $Wn$ | Power scheduled for generator n |
| $x1, x2$ | Power demanded |
| $m1, m2$ | Cost operation results and minimum and maximum power at the loads |
| $\mu 1$ | Represents the ability of the system to adequately manage the optimal power flow to meet the demand |
| $\mu 2$ | Shows that the power grid has mechanisms to minimize mandatory load shedding as the worst case scenario for the network operator and the loads |
| $\mu$ | Quantify respectively a fully resilient and a zero resilient network |
| $V_n$ | Voltage magnitude at bus n |

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
