# Peer review of "Vulnerability Analysis to Maximize the Resilience of Power Systems Considering Demand Response and Distributed Generation"

_electronics, doi:10.3390/electronics10121498_

Round 1

Reviewer 1 Report

This is an interesting issue for power system research. The paper is well written and presented. however, the authors are invited to address the following concerns to reach the publication level:

  1. In the introduction, firstly, more papers should be reviewed from the viewpoint of vulnerability and power system attacks, especially in recent years. Besides, the key contributions of your study should be highlighted in some bullet points in the last paragraph (in comparison to other studies).
  2. Please provide a table of acronyms for further clarification.
  3. In section 2, it is stated that the attacks aim to maximize the operation cost. How these attacks are translated into real measures in the real world? Please clarify in the context.
  4. In section 3.1, the authors used the AC power flow to mitigate the attacks. Is the computation time a key factor in this case? If so, what about fast power flow methods, e.g. DC, fast decoupled, etc? Are they applicable to your approach? Please clarify in the paper.
  5. In section 3.3, a criterion is stressed for the power system resilience. Could you discriminate between resiliency and flexibility in this case? Does the power system use demand flexibility to mitigate attacks?
  6.  The conclusion is supposed to convey the key findings of the research. Moreover, it is a good idea to state the current challenges to pave the way for the next researchers.

Author Response

RESPONSE TO REFEREE 1

Vulnerability Analysis to Maximize the Resilience of Power Systems Considering Demand Response and Distributed Generation

Darin Jairo Mosquera Palacios, Edwin Rivas Trujillo and Jesús M. López-Lezama

June 15th, 2021

We would like to thank the reviewer for his/her valuable comments which have contributed to an improved version of the document. All suggestions have been addressed and track changes have been used to easily identify all changes in the document.

Comment 1:

“In the introduction, firstly, more papers should be reviewed from the viewpoint of vulnerability and power system attacks, especially in recent years. Besides, the key contributions of your study should be highlighted in some bullet points in the last paragraph (in comparison to other studies)”

Response 1:

We improved the introduction section of the paper as suggested by the reviewer. New references were included with their corresponding analysis. Furthermore, the introduction section was divided in three sub-sections, namely: i) motivation, ii) literature review and iii) contribution and paper organization. The new references include works on vulnerability of the last three years and are provided below:    

  1. Yang, S.; Chen, W.; Zhang, X.; Liang, C.; Wang, H.; Cui, W. A Graph-Based Model for Transmission Network Vulnerability Analysis. IEEE Syst. J. 2020, 14, 1447–1456, doi:10.1109/JSYST.2019.2919958.
  2. Liu, B.; Li, Z.; Chen, X.; Huang, Y.; Liu, X. Recognition and Vulnerability Analysis of Key Nodes in Power Grid Based on Complex Network Centrality. IEEE Trans. Circuits Syst. II Express Briefs 2018, 65, 346–350, doi:10.1109/TCSII.2017.2705482.
  3. Beyza, J.; Garcia-Paricio, E.; Ruiz, H.F.; Yusta, J.M. Geodesic Vulnerability Approach for Identification of Critical Buses in Power Systems. J. Mod. Power Syst. Clean Energy 2021, 9, 37–45, doi:10.35833/MPCE.2018.000779.
  4. Luo, L.; Han, B.; Rosas-Casals, M. Network hierarchy evolution and system vulnerability in power grids. IEEE Syst. J. 2018, 12, 2721–2728, doi:10.1109/JSYST.2016.2628410.
  5. Pu, C.; Wu, P.; Xia, Y. Vulnerability Assessment of Power Grids against Link-Based Attacks. IEEE Trans. Circuits Syst. II Express Briefs 2020, 67, 2209–2213, doi:10.1109/TCSII.2019.2958313.
  6. Fang, J.; Su, C.; Chen, Z.; Sun, H.; Lund, P. Power system structural vulnerability assessment based on an improved maximum flow approach. IEEE Trans. Smart Grid 2018, 9, 777–785, doi:10.1109/TSG.2016.2565619.

As regards the key contributions (in comparison with other studies) these were highlighted in bullet points as suggested by the reviewer and are presented below:

  • It complements previous works reported in the specialized literature regarding the solution of the EGIP by considering an AC modelling of the problem as well as simultaneous attacks on lines and generators.
  • New metrics are proposed for the assessment of power system resiliency under deliberate attacks.
  • Enhancement of grid resiliency is proposed by introducing the effect of DERs as a reaction strategy of the system operator.

Comment 2:

“Please provide a table of acronyms for further clarification”

Response 2:

In the new version of the document we included a list of acronyms (labeled as Acronyms and Abbreviations) as suggested by the reviewer, which is shown below. Furthermore, we also included a nomenclature section.   

Acronyms and Abbreviations

AC

Alternating Current

DC

Direct Current

DERs

Distributed Energy Resources

DG

Distributed Generation

DR

Demand Response

EPS

Electric Power Systems

GA

Genetic Algorithm

PSO

Particle Swarm Optimization

Comment 3:

“In section 2, it is stated that the attacks aim to maximize the operation cost. How these attacks are translated into real measures in the real world? Please clarify in the context”

Response 3:

Malicious attacks aim at maximizing damage to the network. Such damage can be ultimately translated into additional costs (repair costs of elements such as lines, transformers, and towers, as well as eventual compensations to consumers due to load shedding). System operators are in charge of assessing these costs as reported in the following reference which was included in the new version of the document.  

Corredor, P.H.; Ruiz, M.E. Mitigating the Impact of Terrorist Activity on Colombia’s Power System. IEEE Power Energy Mag. 2011, 9, 59–66  

Comment 4:

“In section 3.1, the authors used the AC power flow to mitigate the attacks. Is the computation time a key factor in this case? If so, what about fast power flow methods, e.g. DC, fast decoupled, etc? Are they applicable to your approach? Please clarify in the paper”

Response 4:

Considering an AC power flow model allows to approach the problem in its real dimensions. In this particular case, computation time is not a key factor. Further studies may include the use of other models such as the ones suggested by the reviewer. Nonetheless they are currently out of the scope of the present work. We have made mentioned this in the conclusions section as future work as follows:  

Future work may include other power flow models as well as new metrics for resiliency and the use of novel techniques to approach the bi-level programming framework that models the attacker-defender dynamic. A management system that allows the system operator to prioritize critical loads in the event of a deliberate attack may also be implemented. Finally, a future challenge is the implementation of other game-theoretic approaches such as tri-level optimization that also captures the response of the system planner.

Comment 5:

“In section 3.3, a criterion is stressed for the power system resilience. Could you discriminate between resiliency and flexibility in this case? Does the power system use demand flexibility to mitigate attacks?”

Response 5:

Flexibility can be seen as an attribute of a resilient network. In this particular case, the fact of having demand flexibility (demand response) allows the system operator to better handle certain outages; nonetheless, what gives resiliency to the network is to count with a wide set of resources to mitigate eventual attacks that besides demand response include distributed generation and redispatching available resources.

Comment 6:

“The conclusion is supposed to convey the key findings of the research. Moreover, it is a good idea to state the current challenges to pave the way for the next researchers.”

Response 6:

Thank you for the observation. We have included a new paragraph in the conclusion section stating future work and challenges. The new paragraph is provided below:

Future work may include other power flow models as well as new metrics for resiliency and the use of novel techniques to approach the bi-level programming framework that models the attacker-defender dynamic. A management system that allows the system operator to prioritize critical loads in the event of a deliberate attack may also be implemented. Finally, a future challenge is the implementation of other game-theoretic approaches such as tri-level optimization that also captures the response of the system planner.

Reviewer 2 Report

The general idea of the paper seems to be good. However, the paper organization is not acceptable and there are several minor technical challenges that should be effectively addressed.

Comments:

  • There are some grammatical errors and typos that should be corrected before publication.
  • It is recommended to provide a nomenclature at the beginning of the paper to define all variables clearly.
  • Introduction has been vaguely written. My suggestion is to divide the introduction into three subsections: 1) motivation and incitement, 2) literature review and 3) contribution and paper organization.
  • The main contribution of the paper should be highlighted and emphasized. it would be great if the drawbacks and gaps of literature are clear and, particularly, how the proposed approach aims at filling these gaps.
  • The proposed strategy should be compared with other reported strategies.
  • The computation burden and operation time of the proposed method using GA should be presented and compared with other reported techniques.
  • Simulation Results should be enriched. More detailed scenarios should be added.
  • A separate section should be added for discussion of obtained results and main achievements.
  • Quality of figures should be improved.
  • List of abbreviations should be add at the end of the paper

Author Response

RESPONSE TO REFEREE 2

Vulnerability Analysis to Maximize the Resilience of Power Systems Considering Demand Response and Distributed Generation

Darin Jairo Mosquera Palacios, Edwin Rivas Trujillo and Jesús M. López-Lezama

June 15th, 2021

We would like to thank the reviewer for his/her valuable comments which have contributed to an improved version of the document. All suggestions have been addressed and track changes have been used to easily identify all changes in the document.

Comment 1:

“There are some grammatical errors and typos that should be corrected before publication”

Response 1:

Thank you for the observation. We have checked the document and corrected typos and grammar mistakes.  All changes can be verified in tack changes in the new version of the document.

Comment 2:

“It is recommended to provide a nomenclature at the beginning of the paper to define all variables clearly”

Response 2:

We have included a nomenclature, as well as a list of acronyms to define all variables clearly. These are included at the end of the document according to the template of the journal.

Comment 3:

“Introduction has been vaguely written. My suggestion is to divide the introduction into three subsections: 1) motivation and incitement, 2) literature review and 3) contribution and paper organization”

Response 3:

We have improved the introduction by adding the subsections suggested by the reviewer. Furthermore, we have included new references in the literature review. The new references are provided below:

  1. Yang, S.; Chen, W.; Zhang, X.; Liang, C.; Wang, H.; Cui, W. A Graph-Based Model for Transmission Network Vulnerability Analysis. IEEE Syst. J. 2020, 14, 1447–1456, doi:10.1109/JSYST.2019.2919958.
  2. Liu, B.; Li, Z.; Chen, X.; Huang, Y.; Liu, X. Recognition and Vulnerability Analysis of Key Nodes in Power Grid Based on Complex Network Centrality. IEEE Trans. Circuits Syst. II Express Briefs 2018, 65, 346–350, doi:10.1109/TCSII.2017.2705482.
  3. Beyza, J.; Garcia-Paricio, E.; Ruiz, H.F.; Yusta, J.M. Geodesic Vulnerability Approach for Identification of Critical Buses in Power Systems. J. Mod. Power Syst. Clean Energy 2021, 9, 37–45, doi:10.35833/MPCE.2018.000779.
  4. Luo, L.; Han, B.; Rosas-Casals, M. Network hierarchy evolution and system vulnerability in power grids. IEEE Syst. J. 2018, 12, 2721–2728, doi:10.1109/JSYST.2016.2628410.
  5. Pu, C.; Wu, P.; Xia, Y. Vulnerability Assessment of Power Grids against Link-Based Attacks. IEEE Trans. Circuits Syst. II Express Briefs 2020, 67, 2209–2213, doi:10.1109/TCSII.2019.2958313.
  6. Fang, J.; Su, C.; Chen, Z.; Sun, H.; Lund, P. Power system structural vulnerability assessment based on an improved maximum flow approach. IEEE Trans. Smart Grid 2018, 9, 777–785, doi:10.1109/TSG.2016.2565619.

Comment 4:

“The main contribution of the paper should be highlighted and emphasized. it would be great if the drawbacks and gaps of literature are clear and, particularly, how the proposed approach aims at filling these gaps”

Response 4:

Thank you for the observation. In the new version of the document we clearly state the contribution of the paper. These are provided below:

  • It complements previous works reported in the specialized literature regarding the solution of the EGIP by considering an AC modelling of the problem as well as simultaneous attacks on lines and generators.
  • New metrics are proposed for the assessment of power system resiliency under deliberate attacks.
  • Enhancement of grid resiliency is proposed by introducing the effect of DERs as a reaction strategy of the system operator.

Comment 5:

“The proposed strategy should be compared with other reported strategies”

Response 5:

One of the novelties of the proposed approach lies on considering resiliency within a bi-level optimization approach. Also, different costs of load shedding are assigned to critical loads.  To the best knowledge of the authors, similar or comparable approaches have not been reported in the specialized literate limiting an eventual comparison.

Comment 6:

“The computation burden and operation time of the proposed method using GA should be presented and compared with other reported techniques”

Response 6:

Computation times of the simulations were included in the document as instructed by the reviewer. These are included in the discussion section as follows:

For the 5-bus test system the average execution time was 2 minutes and 42 seconds; while for the IEEE RTS 24-bus test system it was 29 minutes and 50 seconds. All tests were carried out on a laptop under windows 10 operating system; with an i7 Pentium Core processor and 8 Ghz of RAM memory.

As regards comparison with other reported techniques please see answer to comment 5.

Comment 7:

“Simulation Results should be enriched. More detailed scenarios should be added.”

Response 7:

Thank you for the observation. In the new version of the document we included a new case study with the IEEE RTS 24-bus test system. Therefore the section of tests and results was complemented with new results and analyses. The new system is shown below.

Comment 8:

“A separate section should be added for discussion of obtained results and main achievements.”

Response 8:

A new section was included for discussion of the obtained results with both test systems.

Comment 9:

“Quality of figures should be improved.”

Response 9:

We have improved the quality of the figures as instructed by the reviewer. Please see the new versions of figures 3, 4, 5 and 6.

Comment 10:

“List of abbreviations should be add at the end of the paper.”

Response 10:

A list of abbreviations and notation was included at the end of the document.

Round 2

Reviewer 1 Report

The authors have addressed all the concerns.